# A Broad-Spectrum Monoclonal Antibody-Based Heterologous ic-ELISA for the Detection of Multiple Pyrethroids in Water, Milk, Celery, and Leek

**DOI:** 10.3390/foods14050768

**Published:** 2025-02-24

**Authors:** Sulin Hou, Dandan Zhang, Zhenyu Xu, Yun Shen, Yulian Wang

**Affiliations:** National Reference Laboratory of Veterinary Drug Residues (HZAU) and MOA Key Laboratory for Detection of Veterinary Drug Residues, Huazhong Agricultural University, Wuhan 430070, China; housulinn@163.com (S.H.); danzhi420@163.com (D.Z.); 2024302010140xzy@webmail.hzau.edu.cn (Z.X.); sy7@webmail.hzau.edu.cn (Y.S.)

**Keywords:** pyrethroids, monoclonal antibody, enzyme-linked immunosorbent assay, milk, celery

## Abstract

Pyrethroids are one of the most commonly used insecticides worldwide in agriculture, public health, and household products. To monitor the presence of pyrethroids in the environment and in food, a broad-spectrum monoclonal antibody (mAb), CL/CN-1D2, was prepared. This mAb demonstrates a 50% inhibitory concentration (IC50) for different pyrethroids: cypermethrin (129.1 µg/L), β-cypermethrin (199.6 µg/L), cyfluthrin (215.5 µg/L), fenpropathrin (220.3 µg/L), λ-cyhalothrin (226.9 µg/L), β-cyfluthrin (241.7 µg/L), deltamethrin (591.2 µg/L), and fenvalerate (763.1 µg/L). Using the mAb CL/CN-1D2, a highly sensitive heterologous indirect competitive ELISA (ic-ELISA) was developed for the rapid detection of these pyrethroids. The limit of detection (LOD) for the eight pyrethroids in water, milk, celery, and leek matrices ranged from 24.4 to 152.2 μg/kg. The recoveries ranged from 65.1% to 112.4%, with a coefficient of variation (CV) below 15%. A robust correlation (*R*^2^ = 0.9945) between the ic-ELISA and GC indicated that the ic-ELISA is a reliable tool for the rapid and cost-effective screening of pyrethroids residues.

## 1. Introduction

Pyrethroids are one of the most commonly used insecticides worldwide, commanding a global market share exceeding 30% [1]. They are widely utilized in agriculture, public health, and household products due to their broad-spectrum efficacy against a variety of pests, including mosquitoes, flies, cockroaches, and agricultural pests such as aphids, caterpillars, and beetles [2,3]. Pyrethroids are classified into type I (e.g., permethrin) and type II (e.g., deltamethrin and cypermethrin) based on the presence of a cyano group. Among these, cypermethrin and deltamethrin are well known for their high insecticidal efficacy. The LD_50_ of cypermethrin in rats is 205 mg/kg [4,5]. Pyrethroids are generally considered to be of low toxicity to mammals, including humans, when used according to guidelines and regulations [6], but their widespread and excessive application can nonetheless pose risks to human health and the environment. Acute exposure to pyrethroids can cause skin irritation, eye irritation, and respiratory distress [7]. Prolonged exposure to pyrethroids has been associated with chronic effects such as neurological damage [8], reproductive disorders [9], hearing loss [10], and developmental abnormalities [11,12,13]. Research has demonstrated that pyrethroids may possess endocrine-disrupting effects, resulting in hormonal imbalances and potential repercussions for reproduction and the immune system [14]. Therefore, it is essential to monitor the presence of pyrethroids in the environment and food to reduce the risk to humans, animals, and the environment.

Currently, pyrethroids are more frequently detected using instrumental methods such as gas chromatography (GC) [15], high-performance liquid chromatography (HPLC) [16], gas chromatography–mass spectrometry (GC-MS) [17,18], and liquid chromatography–mass spectrometry (LC-MS/MS) [19]. These methods are highly sensitive, selective, accurate, and reproducible, enabling the detection of pyrethroids even at very low concentrations. However, they can be time-consuming due to the sample preparation steps such as purification, extraction, and concentration. They require specialized equipment and personnel, which are unsuitable for immediate on-site testing and the initial screening of a large number of samples.

Enzyme-linked immunosorbent assays (ELISAs) use specific antibodies and antigens, proving valuable tools for swift and convenient field testing. Due to their high sensitivity, reliability, cost-effectiveness, ease of use, and portability, ELISAs are widely utilized in food safety and environmental monitoring. In recent years, several polyclonal antibodies (pAbs) and monoclonal antibodies (mAbs) against various pyrethroids have been reported. These include pAbs against λ-cyhalothrin [20,21], deltamethrin, and cypermethrin [22], as well as mAbs against deltamethrin [23]. The currently available detection methods for pyrethroids based on antibodies face several limitations. A significant drawback is their low sensitivity, which can result in the inability to detect pyrethroids when their levels in the environment and foods fall below the detection threshold, making it challenging to assess the risks associated with these chemicals accurately. Another limitation is the narrow detection spectrum of many of these methods, which restricts their ability to identify specific types of pyrethroids. Furthermore, the lack of portability of these methods impedes their application in the field, particularly in remote or inaccessible areas where pyrethroid use is common. Overall, the development of new, more sensitive, and portable methods for detecting pyrethroid residues is essential.

In this study, three haptens of pyrethroids were designed and synthesized to develop highly sensitive, broad-spectrum mAbs capable of detecting eight pyrethroids. Additionally, a heterologous indirect competitive ELISA (ic-ELISA) was established for the detection of these eight pyrethroids in milk, celery, leek, and water with a simplified sample preparation process.

## 2. Materials and Methods

### 2.1. Chemicals

Standard analytes, including cypermethrin, β-cypermethrin, cyfluthrin, β-cyfluthrin, λ-cyhalothrin, deltamethrin, fenpropathrin, and fenvalerate (purity ≥ 99%) were purchased from Dr. Ehrenstorfer (Augsburg, Germany). Bovine serum albumin (BSA), ovalbumin (OVA), keyhole limpet hemocyanin (KLH), Freund’s adjuvants (complete and incomplete), urea hydrogen peroxide, polyethylene glycol 1450 (PEG 1450, 50%), hypoxanthine aminopterin thymidine (HAT), hypoxanthine thymidine (HT), and l-glutamine were purchased from Sigma-Aldrich (St. Louis, MO, USA). 3,3′,5,5′-tetramethyl benzidine (TMB), goat anti-mouse IgG horseradish peroxidase conjugate (HRP-IgG), and HRP-IgG diluent were purchased from Wuhan Feiyi Technology Co., Ltd. (Wuhan, China). Culture media RPMI-1640 and penicillin-streptomycin solution (100×) were purchased from Hyclone (Shanghai, China). Fetal calf serum was purchased from GemCell^TM^ (West Sacramento, CA, USA), and n-hydroxysuccinimide (NHS, chemical purity ≥ 98%), dicyclohexyl-carbodiimide (DCC, chemical purity ≥ 98%), and 1-ethyl-3-(3-dimethylaminopropyl carbodiimide (EDC) were purchased from Sinopharm Chemical Reagent Co., Ltd. (Shanghai, China). Other chemicals were purchased from Sinopharm Chemical Reagent Co., Ltd. (Shanghai, China). All chemicals were of analytical grade unless specified.

### 2.2. Animals and Cells

Female Balb/c mice (6 to 8 weeks old) were obtained from the Hubei Provincial Center for Disease Control and Prevention (Wuhan, China). Mice were housed in pathogen-free conditions with controlled temperature (22 ± 2 °C) and humidity (50 ± 10%) and a 12 h light/dark cycle. All animal experiments were conducted in accordance with the guidelines of the Animal Experiment Center of Huazhong Agricultural University (HZAU) and were approved by the Animal Ethics Committee (No: HZAUMO-2022-0115)

The SP2/0 myeloma cell line was obtained from the National Reference Laboratory of Veterinary Drug Residues (HZAU) (Wuhan, China).

### 2.3. Synthesis of Haptens

Three haptens, namely hapten A, hapten B, and hapten C, were synthesized, with the synthetic routes illustrated in Figure 1. Haptens A, B, and C were synthesized and identified by IT-TOF-MS (Shimadzu Corporation, Kyoto, Japan). The mass spectrometer was operated in positive ion mode with a scan range of *m*/*z* 100–1000. The sample was dissolved in 1 mL of methanol, separated by chromatography, and then injected. The spectra were obtained by scanning in positive/negative ion mode using an ESI source. The detailed procedures for hapten synthesis are outlined as follows.

#### 2.3.1. Synthesis of Hapten A

First, 19.6 g of ethyl tiglate was dissolved in 200 mL of distilled water, followed by the gradual addition of 31.6 g of potassium permanganate (KMnO_4_). The mixture was stirred at 50 °C for 12 h. Subsequently, 50 mL of 5 M sulfuric acid (H_2_SO_4_) was added, and the reaction continued at 60 °C for an additional 6 h. Then, 200 mL of ethyl acetate was added to the mixture, and the organic phase was separated. The aqueous phase was re-extracted, and the organic phases were combined and washed with saturated brine. The combined organic phase was dried overnight with anhydrous sodium sulfate (Na_2_SO_4_) to yield a white solid product.

Subsequently, 7.9 g of the white solid product was dissolved in 100 mL of dichloromethane (CH_2_Cl_2_), followed by the addition of 24.5 g of thionyl chloride (SOCl_2_). The mixture was refluxed at 50 °C for 12 h. After reflux, the solvent and any unreacted thionyl chloride were evaporated under reduced pressure, yielding a residue identified as 3,3-dimethyl-1,2-cyclopropanedicarboxylic anhydride. The anhydride was then dissolved in 50 mL of dichloromethane, and 10 mL of hydroxyethyl solution was added slowly. The reaction proceeded for 30 min, after which 50 mL of saturated ammonium chloride solution and 50 mL of dichloromethane were sequentially added. The mixture was allowed to separate into layers, and the organic phase was collected, washed with saturated brine, and dried over anhydrous sodium sulfate. The product was recovered and purified by column chromatography using a mixture of petroleum ether and ethyl acetate (4:1, *v*/*v*) as the eluent.

#### 2.3.2. Synthesis of Hapten B

To a solution of 19.8 g of m-phenoxybenzaldehyde in 100 mL of anhydrous methanol, 7.6 g of sodium borohydride was added, and the reaction mixture was stirred at room temperature for 2 h. The methanol was then evaporated under reduced pressure, and the residue was redissolved in 50 mL of distilled water. The aqueous solution was extracted twice with 100 mL portions of ethyl acetate. The combined organic layers were washed three times with 50 mL portions of saturated brine, dried over anhydrous sodium sulfate, and filtered. The solvent was removed under reduced pressure to obtain the product as a colorless oil.

Next, 2.0 g of the colorless oily product was dissolved in 50 mL of anhydrous pyridine, and 1.3 g of succinic anhydride was added. The reaction mixture was heated at 90 °C for 12 h. After cooling to room temperature, the mixture was extracted twice with 100 mL portions of ethyl acetate. The combined organic extracts were washed three times with 50 mL portions of saturated brine, dried over anhydrous sodium sulfate, and filtered. The solvent was removed under reduced pressure to give the crude product. Purification by column chromatography on silica gel using a gradient of petroleum ether and ethyl acetate (4:1, *v*/*v*) provided the final product as a colorless oil.

#### 2.3.3. Synthesis of Hapten C

An amount of 2.25 g of (S)-(3-phenoxyphenyl) hydroxyacetonitrile was dissolved in 25 mL of anhydrous pyridine. To this solution, 1.3 g of succinic anhydride was added, and the mixture was allowed to react at 90 °C for 12 h. The progress of the reaction was monitored using TLC, which indicated its completion. Subsequently, 50 mL of 1 M HCl was added, followed by ethyl acetate extraction and three washes with 50 mL of saturated saline solution. The mixture was then dried overnight with anhydrous sodium sulfate. The product was recovered and subsequently purified by column chromatography using a mixture of petroleum ether and ethyl acetate (3:1, *v*/*v*), forming a white solid product.

### 2.4. Synthesis of Antigens

#### 2.4.1. Synthesis of A-DCC-KLH, A-DCC-BSA, and A-DCC-OVA

Firstly, 15 mg of hapten A was dissolved in 1 mL of DMF. To this solution, 10 mg of NHS and 20 mg of DCC were added and stirred overnight at room temperature without light. Subsequently, the mixture was filtered to obtain “solution A”. In a separate step, 10 mg of KLH, 30 mg of BSA, and 36 mg of OVA were each dissolved in 8 mL of PBS, forming “KLH solution B”, “BSA solution B”, and “OVA solution B”, respectively. Solution A was then slowly added to each type of solution B. After the reaction was stirred overnight at 4 °C, A-DCC-KLH, A-DCC-BSA, and A-DCC-OVA were obtained from the solution after dialysis and centrifugation and subsequently stored at −20 °C.

#### 2.4.2. Synthesis of B-DCC-BSA

A total of 15 mg of hapten B was dissolved in 0.7 mL of DMF. To this solution, 28.9 mg of NHS and 51.9 mg of DCC were added and stirred overnight at room temperature without light. After the reaction, it was filtered to obtain “solution A”. Separately, 55.7 mg, 74.3 mg, 111.5 mg, and 222.9 mg of BSA were dissolved in 8 mL of PBS, respectively, forming what we call “solution B”. Solution A was slowly added to solution B. After the reaction, which was stirred overnight at 4 °C, B-DCC-BSA was obtained from the solution after dialysis and centrifugation, and they were stored at −20 °C.

#### 2.4.3. Synthesis of C-DCC-OVA

A total of 19 mg of hapten C was dissolved in 1 mL of DMF. To this solution, 10 mg of NHS and 20 mg of DCC were added and stirred overnight at room temperature. After the reaction, the solution was filtered to obtain “solution A”. Separately, 22 mg of OVA was dissolved in 7 mL of PBS, forming what we call “solution B”. Solution A was slowly added to solution B and stirred overnight at 4 °C. C-DCC-OVA was obtained after dialysis and centrifugation and stored at −20 °C.

The changes in UV absorption for the five antigens were measured to ascertain whether the hapten had bound to the carrier protein. Concurrently, protein concentrations were determined using a BCA kit, and the UV absorption values were employed to calculate the conjugation ratios (Ca/Cb).Ca/Cb = (ACam·KBbm − ACbm·KBam)/(ACbm·Kaam − ACam·KAbm).

### 2.5. Preparation of mAb

A-DCC-KLH and A-DCC-BSA were selected as immunizing antigens, and mice were randomly divided into four groups, each of which was immunized with 50 μg or 100 μg of immunogen. The mouse serum with the highest titer and best specificity was selected for booster injection. Three days after the last immunization, splenocytes were collected from selected mice and fused with SP2/0 myeloma cells in a 5:10 ratio. Six days after cell fusion, hybridoma cells in positive wells were screened using ic-ELISA. They were subsequently subcloned four times using limited dilution to screen for positive hybridoma cells. These positive hybridoma cells were then injected into the peritoneal cavity of female Balb/c mice to generate antibodies, and the mAb was purified by ammonium sulfate precipitation.

### 2.6. ic-ELISA Procedure

The square array titration method was combined with the ic-ELISA technique to determine the type of encapsulated antigen and the optimal conjugation ratio and to optimize the encapsulation concentration and antibody concentration. Finally, the optimal competition time was determined using the ic-ELISA method.

According to the optimized conditions, 100 μL of the coated antigen solution, diluted with CBS, was added to each well and incubated overnight at 4 °C. After washing the plate 3 times with PBST, blocking buffer (250 μL/well) was added and incubated at 37 °C for 2 h. After PBST wash, 50 µL of standard solution or sample and 50 µL of diluted antibody were added to each well. The mixture was incubated at 37 °C for 60 min and then the plate was washed again. Then, 100 µL of goat anti-mouse IgG-HRP diluted in PBS (1:5000) was added to each well and incubated at 37 °C for 30 min. After washing the plate 3 times with PBST, 100 µL of freshly mixed TMB/H_2_O_2_ substrate solution (1:100) was added to each well, and the color was developed for 15 min before adding 50 µL of termination solution/well. Absorbance at 450 nm was measured using an enzyme marker (EnVision, PerkinElmer, Waltham, MA, USA).

### 2.7. Standard Curve and Cross-Reactivity (CR) for ic-ELISA

The standard solution of cypermethrin was diluted to various concentrations (2560, 1280, 640, 320, 160, 80, 40, and 0 µg/L) using PBS. The standard solutions were analyzed using the established ic-ELISA following the optimization conditions and procedure described in Section 2.6. Subsequently, the data were processed using ELISA Calc v 0.1 software to generate the standard curves. From these standard curves, the concentration of cypermethrin that resulted in half-maximal inhibition (IC_50_ value) was calculated. The CR was evaluated by comparing the IC_50_ of cypermethrin to that of its structural analogs, which included β-cypermethrin, cyfluthrin, fenpropathrin, λ-cyhalothrin, β-cyfluthrin, deltamethrin, fenvalerate, esfenvalerate, and permethrin. CR was calculated as follows: CR (%) = IC_50_ (cypermethrin)/IC_50_ (analog) × 100%.

### 2.8. Sample Preparation

Milk, celery, and leek samples were purchased from the educational supermarket of Huazhong Agricultural University in Wuhan, Hubei Province, China (milk was 250 mL of boxed pasteurized pure milk). Water samples were collected from the environmental water of South Lake adjacent to Huazhong Agricultural University in Wuhan, Hubei Province.

Each milk sample (4 mL) was transferred to a 10 mL centrifuged tube to which 1 mL of DMSO was added. The mixture was vigorously vortexed for 5 min and then centrifuged at 8000 r/min for 10 min at 4 °C. The middle layer of the resulting samples was collected for analysis using the ic-ELISA.

Each water sample (4 mL) was transferred to a 10-mL centrifuged tube, and 1 mL of DMSO was added. The mixture was vortexed for 5 min, followed by centrifuging at 8000 r/min for 10 min at 4 °C. The resulting supernatants were then collected for analysis using the ic-ELISA.

Celery and leek samples were purchased from the vegetable market (Wuhan, Hubei, China). They were minced and homogenized. An amount of 10 g of each homogenized sample was weighed and placed in a 50 mL centrifuged tube. Subsequently, 10 mL of acetonitrile was added. After vortexing for 5 min, 1 g of NaCl and 4 g of MgSO_4_ were added. The mixture was vortexed for an additional 5 min. Following this, the mixture was centrifuged at 6000 r/min for 5 min at 4 °C. The resulting supernatants (1 mL) were carefully collected and evaporated at 40 °C under a stream of nitrogen. The residue was then reconstituted in 1 mL of PBS containing 20% DMSO for ic-ELISA analysis.

### 2.9. Validation of the ic-ELISA

The ic-ELISA was validated using twenty distinct blank samples, including milk, water, celery, and leek, sourced from various origins. All samples were confirmed by gas chromatography (GC) to be free of cypermethrin, β-cypermethrin, cyfluthrin, β-cyfluthrin, λ-cyhalothrin, deltamethrin, fenpropathrin, and fenvalerate. The GC method was referenced from GB/T 5750.9-2006 with certain optimizations. The analysis was performed using an Agilent 7890B gas chromatograph (Agilent Technologies, Santa Clara, CA, USA) equipped with an Agilent DB-1701 capillary column (30 m × 0.25 mm, 0.25 μm; Agilent Technologies, USA). N2 was used as the carrier gas at a flow rate of 50 mL/min. The column temperature was maintained at 240 °C. The injection volume was 1 μL, and the injection mode was set to splitless injection.

Twenty blank samples from different sources were collected and processed using the ic-ELISA method as previously detailed. The average concentration (C: the mean detected concentration of pesticide in the twenty blank samples) and standard deviation (SD: the standard deviation of the detected pesticide concentration in the twenty blank samples) were calculated. The lowest detection limit (LOD) and the lowest quantitative limit (LOQ) were calculated as follows: LOD = C + 3 × SD, LOQ = C + 10 × SD.

The recovery and accuracy of the optimized method were validated using spiked pesticide in blank samples at low, medium, and high levels (1 × LOQ, 2 × LOQ, 4 × LOQ), as well as the concentration equivalent to the maximum residue limits (MRLs). The following equation was employed to assess the recovery rate at each level: Recovery (%) = (detected concentration/spiked concentration) × 100. Furthermore, the coefficient of variation (CV) was calculated by analyzing spiked samples at different levels and conducting five repeated trials.

The reliability of the ic-ELISA was evaluated by comparing it to the GC for analysis of incurred samples. Five blind water samples were detected using both ic-ELISA and GC methods. The correlation between the ic-ELISA and the GC analyses was calculated to evaluate the agreement between the two methods.

## 3. Results and Discussion

### 3.1. Design and Characterization of Haptens

In this study, our objective was to develop monoclonal antibodies with enhanced sensitivity for recognizing a broader spectrum of pyrethroids (Figure 2B). Upon comparing the lowest energy conformer structures of these pyrethroids, it became evident that the primary distinction lay in the substituent on the terminal cyclopropane (Figure 2B). Therefore, three haptens were designed to retain the structural elements of cyclopropane, -CN, and m-phenoxybenzyl in pyrethroids while eliminating the halogen elements. Haptens A, B, and C were synthesized and identified by IT-TOF (Figure 3). The MS *m*/*z* of hapten A was calculated for C_21_H_19_NO_5_ [M + Na]^+^ 388.3840, and it was found to be 388.1120. For hapten B, the MS *m*/*z* was calculated for C_17_H_16_O_5_ [M − H]^−^ 299.092, and it was found to be 299.075. The MS *m*/*z* of hapten C was calculated for C_18_H_15_NO_5_ [M + Na]^+^ 348.34, and it was found to be 348.0802. These results confirm the successful synthesis of haptens A, B, and C.

The lowest energy conformers of the three haptens were overlaid with cypermethrin using SYBYL-X 2.0 (Figure 4A–C). It was evident that hapten A and cypermethrin exhibited a nearly perfect overlap, with only slight differences in their angular positions. Furthermore, these conformational changes had a noticeable effect on the electronic distribution. Hence, the atomic charges of cypermethrin and haptens A, B, and C were analyzed using Chem 3D 2021 software. Obviously, the spatial conformation of hapten A was more like that of cypermethrin rather than that of haptens B and C (Figure 4D). The introduction of the spacer arm notably altered the atomic charge distribution at C, with a difference in atomic charge exceeding 0.3 A.U, while negligible variances were observed at other sites, where the differences were less than 0.1 A.U. Consequently, hapten A was selected as the immunogen to enhance the production of high-affinity antibodies.

Pyrethroids are small molecular compounds that lack intrinsic immunogenic properties and cannot be directly linked to proteins. Studies have indicated that the design of haptens can significantly influence the specificity of the antibody response [24]. Haptens are more likely to elicit immune responses when they are structurally similar to the pesticide and located in the portion of the molecule farther from the carrier protein [25]. Therefore, it is essential to design haptens based on the structure of pyrethroids to enable effective coupling with proteins and induce an immune response.

**Figure 2 foods-14-00768-f002:**
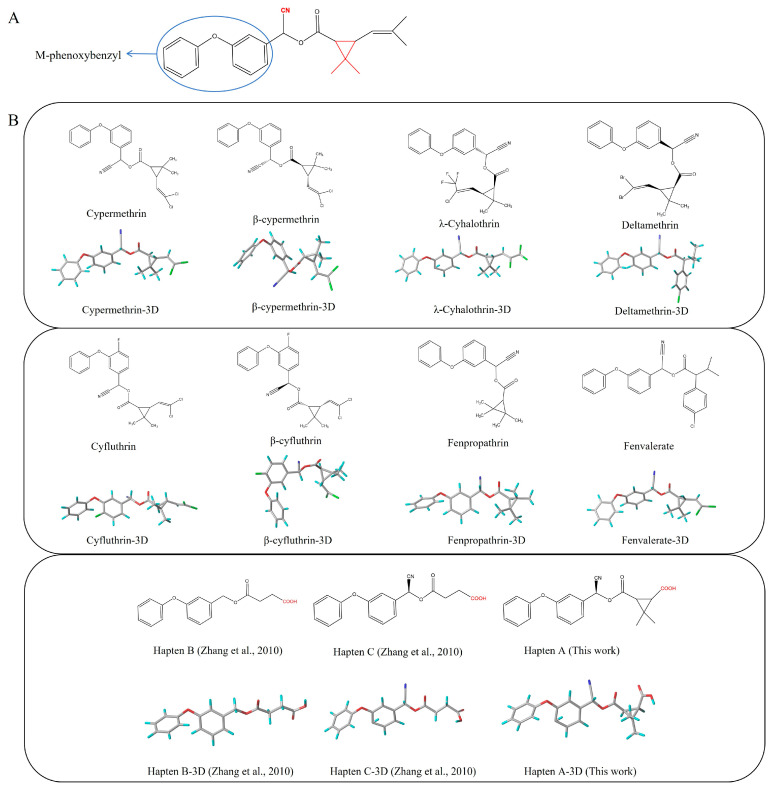
Functional groups of pyrethroids (**A**). The chemical structure and lowest energy conformational structures of cypermethrin, β-cypermethrin, λ-cypermethrin, cypermethrin, cyfluthrin, β-fluorocypermethrin, deltamethrin, hapten B, hapten C, and hapten A. The lowest energy conformational structures are colored as follows: red for oxygen, blue for nitrogen, gray for carbon, and green for chlorine (**B**) [26].

Several haptens derived from various pyrethroids have been reported [20,21,23,26]. The use of the phenoxybenzyl moiety as a hapten structure is limited as it may only enable the recognition of phenoxybenzoic acid and 4-fluoro-3-phenoxy benzoic acid without providing adequate specificity for pyrethroids [26]. Hapten C, used in this study, has been used as an immunogen to prepare mAb in a previous report, which contains -CN and no cyclopropane. The mAb can effectively recognize phenothrin, permethrin, deltamethrin, cypermethrin, and cyhalothrin [27]. A new hapten, which selectively hydrolyzed the “-CN” group of λ-cyhalothrin while preserving the majority of the functional groups, was used to develop the pAb, which exhibited high sensitivity to λ-cyhalothrin but not to the other analogs [20]. When the haptens included -CN, m-phenoxybenzyl, cyclopropane, and the halogen elements, the antibodies were only sensitive to specified pyrethroids, such as cyhalothrin [21] and deltamethrin [23]. The cyclopropane ring and -CN group (Figure 2A) are essential for hydrophobic and dipole interactions with the antibody’s binding pocket. The m-phenoxybenzyl moiety enhances π-π stacking, stabilizing antigen–antibody complexes. In summary, cyclopropane, -CN, and m-phenoxybenzyl are crucial epitopes in the antigen–antibody reaction (Figure 2A), and halogen elements play a significant role in influencing the specificity of antibodies.

**Figure 3 foods-14-00768-f003:**
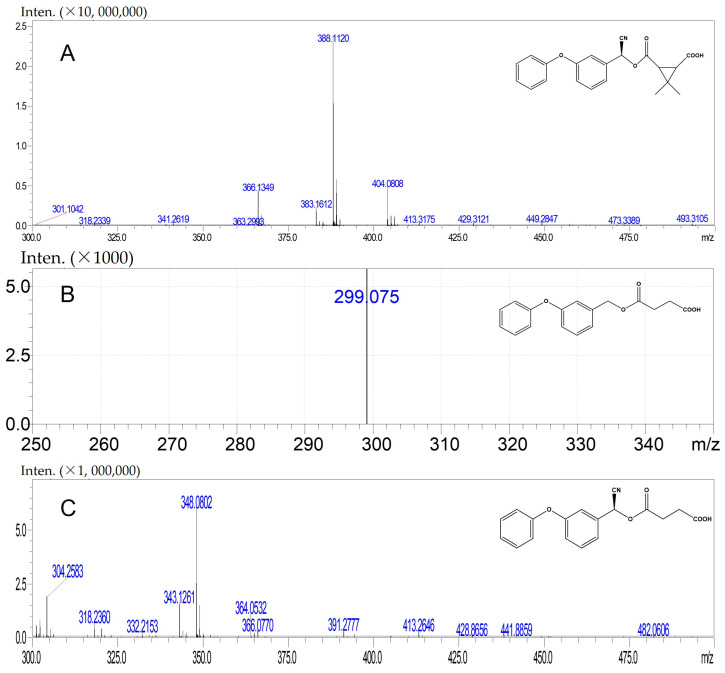
MS spectra of hapten A (**A**), hapten B (**B**), and hapten C (**C**).

**Figure 4 foods-14-00768-f004:**
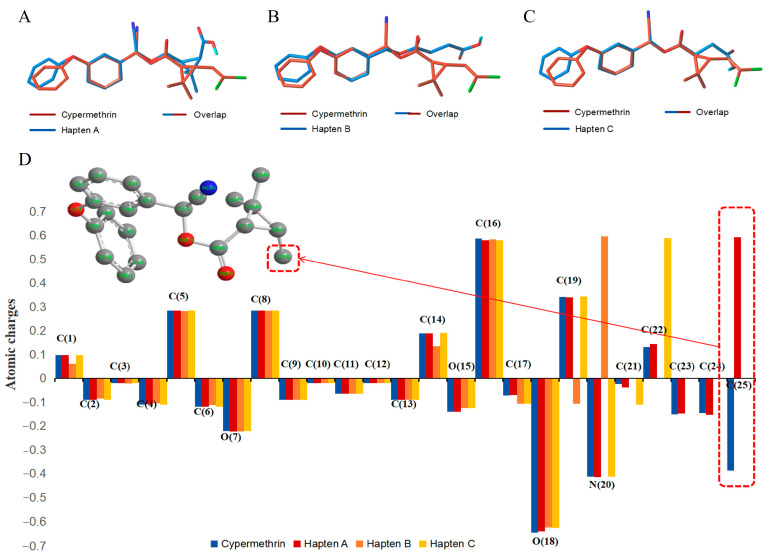
Comparison of simulated cypermethrin and hapten A (**A**), cypermethrin and hapten B (**B**), and cypermethrin and hapten C (**C**). Calculated atomic charges of cypermethrin, hapten A, hapten B, and hapten C (**D**).

### 3.2. Characterization of Antigens

After coupling haptens A, B, and C with proteins to synthesize antigens A-DCC-KLH, A-DCC-BSA, A-DCC-OVA, B-DCC-BSA, and C-DCC-OVA, the antigen, haptens, and proteins were identified using a UV–visible spectrophotometer. The UV absorbance spectra of A-DCC-KLH (λmax, 248 nm), A-DCC-BSA (λmax, 277 nm), A-DCC-OVA (λmax, 258 nm), B-DCC-BSA (λmax, 278 nm), and C-DCC-OVA (λmax, 343 nm) were different from those of BSA (λmax, 279 nm), KLH (λmax, 279 nm), and OVA (λmax, 279 nm). The UV peak shapes also exhibited distinctions among hapten A, hapten B, and hapten C (Appendix A), indicating the successful synthesis of the antigens. The protein concentrations for A-DCC-KLH, A-DCC-BSA, and A-DCC-OVA were 2.2, 3.6, and 2.6 mg/mL, respectively, with conjugation ratios of 6.5, 16.0, and 16.2. For B-DCC-BSA, the protein concentrations were 2.1, 3.9, 1.3, and 3.7 mg/mL, along with conjugation ratios of 13.4, 17.4, 15.3, and 16.0. Additionally, the protein concentration for C-DCC-OVA was 7.8 mg/mL with a conjugation ratio of 21.2.

### 3.3. Identification of Antiserum and mAb

Following the third immunization process, the titer and specificity of the antisera from the immunized mice were characterized using both an indirect ELISA and an ic-ELISA, respectively. The antisera from the mice immunized with A-DCC-KLH at a dose of 100 µg exhibited the highest titer and superior specificity against cypermethrin (Table 1). The mouse with the most robust immune response was chosen for cell fusion. After screening and culturing, the cell line CL-CN/1D2 was selected for mAb production, exhibiting an inhibition rate of 67.7%. The mAb was identified as belonging to the IgG1 subtype with a kappa light chain (Appendix A).

### 3.4. Development and Optimization of ic-ELISA

It has been reported that variations in sensitivity and specificity can arise from the use of homologous and heterologous coating antigens in ELISA [23]. In this study, we investigated the sensitivity and specificity of homologous coating antigens (A-DCC-BSA and A-DCC-OVA) and heterologous coating antigens (B-DCC-BSA and C-DCC-OVA) in ic-ELISA. The antisera from mice immunized with immunogen A-DCC-KLH were assessed using both an indirect ELISA and an ic-ELISA with the four coating antigens. The results indicate that the highest titer was observed for B-DCC-BSA, demonstrating superior specificity against cypermethrin (Table 2). Therefore, heterologous coating (B-DCC-BSA) was chosen in this study.

The sensitivity of ic-ELISA can also be influenced by coating antigens with different conjugation ratios. In this study, we synthesized the coating antigen B-DCC-BSA with four distinct conjugation ratios (13.4, 15.3, 16.0, and 17.4) to systematically evaluate their impacts on sensitivity. Following the square matrix titration procedure, the optimal dilution ratios for mAb CL-CN/1D2 (ascites) and the coating antigen (B_3_-DCC-BSA with a conjugation ratio of 15.3) in heterologous ic-ELISA were 1:5000 and 1.9 µg/mL, respectively (Appendix A).

The optimization of the competitive time was carried out under the specified optimized conditions and in accordance with the ic-ELISA procedure detailed in Section 2.7. As a result, the highest sensitivity was attained at a competitive time of 60 min (Appendix A).

### 3.5. The Standard Curve and the CR for the ic-ELISA

Under optimal conditions, various cypermethrin standard solutions were analyzed. Subsequently, the acquired data were processed using ELISA Calc software to construct the standard curve. The calibration of the cypermthrin solution spanned a concentration ranging from 40 to 1280 µg/L (Figure 5A).

The IC_50_ values for various pyrethroids, including cypermethrin, β-cypermethrin, cyfluthrin, fenpropathrin, λ-cyhalothrin, β-cyfluthrin, deltamethrin, and fenvalerate, were 129.1, 199.6, 215.5, 220.3, 226.9, 241.7, 591.2, and 763.1 μg/L, respectively (Table 3). The CR of mAb CL-CN/1D2 to these pyrethroids ranged from 100.0% for cypermethrin to 16.9% for fenvalerate. Notably, in comparison to previously reported antibodies, the antibody developed in this study exhibited heightened sensitivity and the capability to recognize eight different pyrethroids, most of which exhibited cross-reactivity rates exceeding 50%.

### 3.6. Validation of the ic-ELISA Method

The LODs and LOQs for eight pyrethroids in various matrices, including water, milk, celery, and leek samples, ranged from 24.4 to 152.2 μg/kg and 36.6 to 188.6 μg/kg, respectively (Table 4). The Chinese National Food Safety Standard (GB 2763-2021) specifies the maximum residue limits (MRLs) for pyrethroid pesticides in food [29]. Specifically, the MRLs for cypermethrin in leek and celery are set at 1 mg/kg, while those for deltamethrin range from 0.2 to 2 mg/kg. Similarly, the European Union has established MRLs for pesticide residues in food and feed [30], with cypermethrin in leek and celery set at 1–2 mg/kg and deltamethrin at 0.2–0.5 mg/kg. The ic-ELISA method developed in this study meets the detection requirements for these MRLs and is suitable for the rapid screening of type II pyrethroid pesticides. Using the established heterologous ic-ELISA method, an analysis was conducted on water, milk, celery, and leek samples spiked with varying concentrations (1 × LOQ, 2 × LOQ, 4 × LOQ, MRL) of the eight pyrethroids. The average recovery rates for pyrethroids in these samples ranged from 65.1% to 112.4%. Furthermore, the CV within and between batches was less than 15% (Table 4).

The ic-ELISA was validated by GC through the analysis of five water samples. The ic-ELISA results show good consistency (*R*^2^ = 0.9945) with the results obtained by GC (Figure 5B). This indicates that the established ic-ELISA has high accuracy for the detection of pyrethroids, making it a reliable method for detecting these pyrethroids residues. Although the limits of detection (LODs) of ic-ELISA (24.4–152.2 μg/kg) are higher than those of GC-MS (typically < 10 μg/kg), ic-ELISA offers significant advantages in rapid on-site detection scenarios, such as the preliminary monitoring of agricultural runoff or routine food safety inspections of vegetables.

## 4. Conclusions

In the present work, the synthesis of three haptens and five antigens was accomplished. Utilizing the DCC method, hapten A was conjugated with KLH as the immunogen for the production of mAb against pyrethroids, while hapten B was coupled with BSA to function as the coating antigen. A heterologous ic-ELISA was established based on mAb CL-CN/1D2, yielding an IC_50_ of 129.1 ± 2.7 µg/L for cypermethrin. The cross-reactivity of the assay with various pyrethroids was as follows: cypermethrin (100.0%), β-cypermethrin (64.7%), cyfluthrin (59.9%), fenpropathrin (58.6%), λ-cyhalothrin (56.9%), β-cyfluthrin (53.4%), deltamethrin (21.8%), and fenvalerate (16.9%). The ic-ELISA method screened eight pyrethroids in various samples, including milk, water, celery, and leek, with average recovery rates ranging from 65.1% to 112.4%.

## Figures and Tables

**Figure 1 foods-14-00768-f001:**
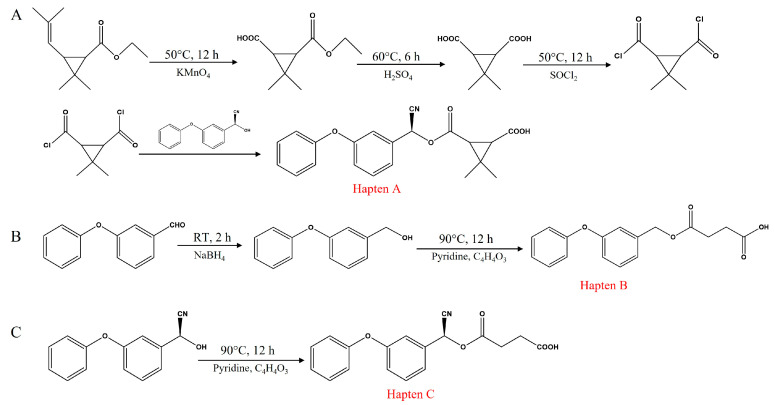
Synthesis routes of haptens (**A**–**C**).

**Figure 5 foods-14-00768-f005:**
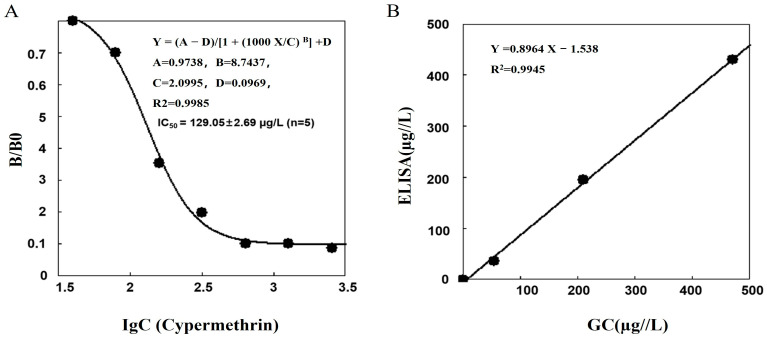
The standard inhibition curve of cypermethrin measured by heterologous ic-ELISA (**A**). The correlation of the cypermethrin assay between the ic-ELISA and the GC method for the incurred water samples (**B**).

**Table 1 foods-14-00768-t001:** The titer and specificity of the antiserum and the specificity of the hybridomas.

Immunogen	No.	Dose (μg)	Titer (1: X ^1^)				(1 − B/B0) % ^2^			
			1	2	3	4	1	2	3	4
A-DCC-KLH	4	50	16,000	16,000	16,000	16,000	48	49	47	50
A-DCC-KLH	4	100	16,000	16,000	32,000	16,000	49	53	50	51
A-DCC-BSA	4	50	16,000	16,000	8000	8000	5	4	7	8
A-DCC-BSA	4	100	16,000	16,000	8000	8000	16	8	17	12

Note: ^1^ “X” represents the dilution factor. ^2^ Cypermethrin = 1000 μg/L.

**Table 2 foods-14-00768-t002:** The optimal screening of coating antigens.

Coating Antigen	Antiserum 1		Antiserum 2		Antiserum 3		Antiserum 4	
	Titer	(1 − B/B0)% ^1^	Titer	(1 − B/B0)% ^1^	Titer	(1 − B/B0)% ^1^	Titer	(1 − B/B0)% ^1^
A-DCC-BSA	16,000	15	16,000	11	16,000	22	16,000	20
A-DCC-OVA	8000	22	8000	32	8000	7	4000	5
B-DCC-BSA	16,000	48	16,000	50	32,000	58	16,000	51
C-DCC-OVA	800	15	800	10	4000	3	4000	2

Note: ^1^ Cypermethrin = 1000 µg/L.

**Table 3 foods-14-00768-t003:** Cross-reactivity rates of various antibodies against pyrethroids [20,21,23,26,28].

Hapten	Antibody	Analyte	IC50 (μg/L)	CR (%)
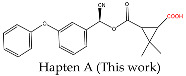	mAb	Cypermethrin	129.1	100.0
		β-Cypermethrin	199.6	64.7
		Cyfluthrin	215.5	59.9
		Fenpropathrin	220.3	58.6
		λ-Cyhalothrin	226.9	56.9
		β-Cyfluthrin	241.7	53.4
		Deltamethrin	591.2	21.8
		Fenvalerate	763.1	16.9
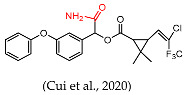	pAb	λ-Cyhalothrin	40.1	100.0
		Deltamethrin	802.6	5.0
		Cypermethrin	802.6	5.0
		Cyfluthrin	401.3	10.0
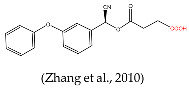	mAb	Phenothrin	204.0	100.0
		Permethrin	325.0	62.8
		Deltamethrin	52.0	392.3
		Cypermethrin	49.0	416.3
		Cyhalothrin	58.0	351.7
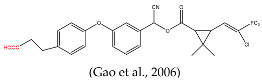	pAb	Cyhalothrin	37.2	100.0
		Cypermethrin	1488.0	2.5
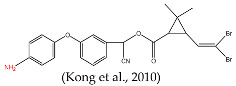	mAb	Deltamethrin	17.0	100.0
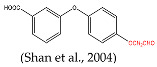	pAb	Phenoxybenzoic acid	1.7	100.0
		4-Fluoro-3-phenoxybenzoic acid	2.3	72.0

**Table 4 foods-14-00768-t004:** The LODs, LOQs, recoveries and CVs (%) of the ic-ELISA for pyrethroids detection in water, milk, celery, and leek samples.

Samples	Compounds	LOD (μg/kg)	LOQ (μg/kg)	Spiked Level (μg/kg)	Recovery (%)	CV (%)
Water	Cypermethrin	25.2	44.4	40, 80, 160	77.4–96.6	6.1–9.8
	β-Cypermethrin	24.4	36.6	40, 80, 160	90.5–102.7	5.5–12.7
	Cyfluthrin	36.0	54.7	60, 120, 240	92.7–101.8	9.0–11.0
	λ-Cyhalothrin	43.2	55.8	50, 100, 200	92.7–103.0	5.0–13.1
	β-Cyfluthrin,	46.0	58.3	50, 100, 200	74.1–105.4	3.9–14.8
	Fenpropathrin	40.0	56.1	50, 100, 200	93.3–106.7	3.0–14.8
	Deltamethrin	25.2	44.4	75, 150, 300	89.8–102.3	5.8–11.5
	Fenvalerate	45.7	75.1	75, 150, 300	81.9–91.0	5.3–7.3
Milk	Cypermethrin	37.5	53.3	50(MRL), 100, 200	82.8–104.2	4.3–14.0
	β-Cypermethrin	37.5	52.8	50(MRL), 100, 200	103.0–106.2	4.0–12.4
	Cyfluthrin	59.0	87.0	80, 160, 320	84.9–96.6	3.8–10.2
	λ-Cyhalothrin	52.7	82.8	80, 160, 200(MRL), 320	83.9–100.5	4.5–12.3
	β-Cyfluthrin,	57.4	80.6	80, 160, 320	79.4–102.6	6.7–12.5
	Fenpropathrin	52.0	81.0	80, 160, 320	66.4–92.5	5.5–10.8
	Deltamethrin	67.4	95.4	100, 200, 400	96.3–103.8	4.6–10.5
	Fenvalerate	68.9	96.9	100(MRL), 200, 400	86.4–103.7	6.2–12.3
Celery	Cypermethrin	68.5	97.2	100, 200, 400, 1000(MRL)	75.7–102.3	4.6–13.1
	β-Cypermethrin	65.5	102.6	100, 200, 400, 1000(MRL)	77.9–99.9	3.4–9.6
	Cyfluthrin	68.6	97.3	100, 200, 400, 500(MRL)	82.9–108.5	2.9–11.4
	λ-Cyhalothrin	72.1	104.3	100, 200, 400, 500(MRL)	88.0–102.5	1.1–3.7
	β-Cyfluthrin,	72.7	102.1	100, 200, 400, 500(MRL)	79.1–92.0	3.3–7.9
	Fenpropathrin	64.1	96.3	100, 200, 400, 1000(MRL)	72.6–93.5	3.9–5.1
	Deltamethrin	143.6	183.5	200, 400, 800, 2000	72.9–99.3	5.4–13.4
	Fenvalerate	148.4	185.5	200, 400, 800	93.0–98.8	2.1–8.6
Leek	Cypermethrin	72.2	99.5	100, 200, 400, 1000(MRL)	74.9–107.4	4.6–10.6
	β-Cypermethrin	67.4	93.3	100, 200, 400, 1000(MRL)	78.7–112.4	4.5–9.4
	Cyfluthrin	64.1	95.6	100, 200, 400, 500(MRL)	65.1–108.2	4.2–7.9
	λ-Cyhalothrin	75.8	105.2	100, 200, 400, 500(MRL)	85.9–100.5	3.2–10.8
	β-Cyfluthrin,	67.3	100.9	100, 200, 400, 500(MRL)	73.1–106.2	5.2–8.1
	Fenpropathrin	65.6	97.1	100, 200, 400, 1000(MRL)	71.9–99.2	2.6–8.9
	Deltamethrin	152.2	188.6	200, 400, 800	92.3–97.9	3.5–6.6
	Fenvalerate	142.1	181.5	200, 400, 800	94.6–105.7	1.5–7.4

## Data Availability

The original contributions presented in this study are included in the article/Appendix A. Further inquiries can be directed to the corresponding author.

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
