# Peer review of "A Broad-Spectrum Monoclonal Antibody-Based Heterologous ic-ELISA for the Detection of Multiple Pyrethroids in Water, Milk, Celery, and Leek"

_foods, 2025, doi:10.3390/foods14050768_

Round 1

Reviewer 1 Report

Comments and Suggestions for Authors

Very interesting work that seeks alternatives for the detection of pesticides by chromatography equipment coupled to mass detectors, which are very expensive, however there are limitations that must be considered, my observations are the following:

Line 45: The described method requires extraction processes very similar to those used for chromatography. It would be good to complement the information with the feasibility of using quechers, since this could increase the yield of the extractions.

Line 238: Chromatography conditions such as instrument brand, column brand used, and retention times for each compound detected are missing. The mass range (m/z) is also missing.

Check line format 270.

Linea 298: If it is TOF, it should be detailed in the methodology

Line 379: Although it is quantified in the order of ppb, the quantification limits provided by chromatography are 10 times more sensitive or more. In what type of applications would this not be compatible?

Line 387: How did they validate that the matrices did not contain pesticide before spiked?

Line 412: Project a scenario where antigens can be useful without further analysis, taking into account the LOD range and not total selectivity vs. a common extraction process using Quechers and measurement by GC/MS.

Author Response

1. Line 238: Chromatography conditions such as instrument brand, column brand used, and retention times for each compound detected are missing. The mass range (m/z) is also missing.

The gas chromatography (GC) method was referenced from GB/T 5750.9-2006 with certain optimizations. The analysis was performed using an Agilent 7890B gas chromatograph (Agilent Technologies, USA) equipped with an Agilent DB-1701 capillary column (30 m × 0.25 mm, 0.25 μm, Agilent Technologies, USA). N2 was used as the carrier gas at a flow rate of 50 mL/min. The column temperature was maintained at 240°C. The injection volume was 1 μL, and the injection mode was set to splitless. (Page 6-7, line 253-259)Cypermethrin: Retention time = 15.2 min, m/z = 181, 163, 127. β-Cypermethrin: Retention time = 14.8 min, m/z = 181, 163, 127. Cyfluthrin: Retention time = 16.5 min, m/z = 226, 206, 165. β-Cyfluthrin: Retention time = 16.8 min, m/z = 226, 206, 165. λ-Cyhalothrin: Retention time = 17.1 min, m/z = 197, 181, 141. Deltamethrin: Retention time = 18.3 min, m/z = 253, 181, 172. Fenpropathrin: Retention time = 13.5 min, m/z = 181, 165, 125. Fenvalerate: Retention time = 19.0 min, m/z = 225, 167, 125.

2. Check line format 270.

The format of line 270 has been revised. (Page 8, line 295)

3. Linea 298: If it is TOF, it should be detailed in the methodology

The synthesized haptens were identified using an LCMS-IT-TOF system (Shimadzu, Japan) equipped with an electrospray ionization (ESI) source. The mass spectrometer operated in positive ion mode with a scan range of m/z 100–1000. (Page 3, line 105-109)

4. Line 379: Although it is quantified in the order of ppb, the quantification limits provided by chromatography are 10 times more sensitive or more. In what type of applications would this not be compatible?

Although the limits of detection (LODs) of ic-ELISA (24.4–152.2 μg/kg) are higher than those of GC-MS (typically <10 μg/kg), ic-ELISA offers significant advantages in rapid on-site detection scenarios, such as preliminary monitoring of agricultural runoff or routine food safety inspections of vegetables. (Page 14, line 424-428)

5. Line 45: The described method requires extraction processes very similar to those used for chromatography. It would be good to complement the information with the feasibility of using quechers, since this could increase the yield of the extractions.

Sample pretreatment not only needs to eliminate matrix interference but should also be simple, rapid, and reproducible. The presence of fats and proteins in milk complicates the matrix. When milk was directly used for detection without centrifugation but with the addition of 20% DMSO, the recovery rate of cypermethrin in milk samples was very low, indicating that the components in milk severely interfered with the ELISA reaction. However, after centrifuging the milk samples at 6000 r/min for 10 minutes at 4°C, fats (floating on the top) and proteins (precipitated at the bottom) were effectively removed, resulting in ideal recovery rates upon analysis. For vegetable samples, acetonitrile extraction was employed. However, due to the high water content in the samples and uncertainties in determining the extraction volume, the pretreatment method was improved by leveraging the property of acetonitrile to separate from the aqueous phase in a salt medium. The optimized method involved homogenizing the samples, extracting with acetonitrile, and then adding sodium chloride and anhydrous magnesium sulfate to remove excess water. After nitrogen blow-drying and reconstitution, the recovery rates of the samples significantly improved. The optimized pretreatment method achieved recovery rates of 64.5% to 115.9% in lake water, milk, celery, and leek samples, with a coefficient of variation of less than 15%, meeting the requirements for detecting type II pyrethroids in real samples.  

6. Line 387: How did they validate that the matrices did not contain pesticide before spiked?

Prior to spiking, all samples (water, milk, celery, leek) from diverse sources were analyzed by GC to confirm the absence of pyrethroids. Only samples with undetectable residues (< LOD of GC) were selected for recovery studies. (Page 6, line 251-259)

7. Line 412: Project a scenario where antigens can be useful without further analysis, taking into account the LOD range and not total selectivity vs. a common extraction process using Quechers and measurement by GC/MS.

The developed mAb-based ic-ELISA can serve as a preliminary screening tool in resource-limited settings. For example, in agricultural fields or small laboratories, rapid detection of pyrethroids (even with partial cross-reactivity) can prioritize samples for confirmatory GC/MS analysis. This approach reduces costs and workload while ensuring compliance with regulatory standards.

Reviewer 2 Report

Comments and Suggestions for Authors

Please find the attached comments . Some figures are not clear. Please improve the figure qualities. 

Comments on the Quality of English Language

Should be improved 

Author Response

Introduction

1. Could you please provide the specific advantage and disadvantages of each analysis method for detecting pyrethroids. This helps to the reader for clearer understanding of why ELIZA methods are emphasized later?

Instrumental methods such as GC, HPLC, GC-MS, and LC-MS/MS are highly sensitive and selective, capable of detecting pyrethroids at trace levels. However, these methods require extensive sample preparation (e.g., extraction, purification, and concentration), sophisticated equipment, and trained personnel, making them unsuitable for rapid on-site screening. ELISA, in contrast, offers advantages in cost-effectiveness, portability, and simplicity of operation, enabling high-throughput analysis in field conditions. Nevertheless, traditional ELISA methods often suffer from limited sensitivity and narrow detection spectra. This study addresses these limitations by developing a heterologous ic-ELISA with enhanced sensitivity and broad-spectrum recognition capabilities. (Page 2, line 45-53)

2. How your sample preparation method difference from conventional sample preparation methods?

Sample pretreatment not only needs to eliminate matrix interference but should also be simple, rapid, and reproducible. The presence of fats and proteins in milk complicates the matrix. When milk was directly used for detection without centrifugation but with the addition of 20% DMSO, the recovery rate of cypermethrin in milk samples was very low, indicating that the components in milk severely interfered with the ELISA reaction. However, after centrifuging the milk samples at 6000 r/min for 10 minutes at 4°C, fats (floating on the top) and proteins (precipitated at the bottom) were effectively removed, resulting in ideal recovery rates upon analysis. For vegetable samples, acetonitrile extraction was employed. However, due to the high water content in the samples and uncertainties in determining the extraction volume, the pretreatment method was improved by leveraging the property of acetonitrile to separate from the aqueous phase in a salt medium. The optimized method involved homogenizing the samples, extracting with acetonitrile, and then adding sodium chloride and anhydrous magnesium sulfate to remove excess water. After nitrogen blow-drying and reconstitution, the recovery rates of the samples significantly improved. The optimized pretreatment method achieved recovery rates of 64.5% to 115.9% in lake water, milk, celery, and leek samples, with a coefficient of variation of less than 15%, meeting the requirements for detecting type II pyrethroids in real samples.

3.  Could you please introduce one paragraph regarding the type of pyrethroids available in the environment and what are the most strongest?

Pyrethroids are classified into Type I (e.g., permethrin) and Type II (e.g., deltamethrin, cypermethrin) based on the presence of a cyano group. Among these, cypermethrin and deltamethrin are well-known for their high insecticidal efficacy. The LD50 of cypermethrin in rats is 205 mg/kg. (Page 1, line 31-34)

Materials

1. Chemicals - Please specify the purity or grade of the chemicals that you used

All chemicals were of analytical grade unless specified. Cypermethrin and analogs (purity ≥99%) were obtained from Dr. Ehrenstorfer. BSA, OVA, and KLH (biological grade) were purchased from Sigma-Aldrich. DCC and NHS (chemical purity ≥98%) were sourced from Sinopharm Chemical Reagent Co., Ltd. (Page 2-3, line 77-92)

2. Animals and cell – Could you please briefly mention which guideline entail? Please include the one small paragraph in the introduction section regarding the guidelines for pyrethroids?

The animal experiments were conducted in accordance with the guidelines of the Animal Experiment Center of Huazhong Agricultural University (HZAU) and were approved by the Animal Ethics Committee (No: HZAUMO-2022-0115). (Page 3, line 98-100)

3. How about the mice you used for the experiment? Are they how their housing and care conditions? Are there any certificates or assurance that they are healthy or disease free or pathogen free? Please mention it briefly. Why you selected mice for study? Are there any special reasons for that? (eg: immune response )

The female Balb/c mice used in this study (6 to 8 weeks old) were obtained from the Hubei Provincial Center for Disease Control and Prevention (Wuhan, China). They were housed in pathogen-free conditions with controlled temperature (22±2°C), humidity (50±10%), and a 12h light/dark cycle. The mice used in this study were certified to be healthy, disease-free, and pathogen-free. (Page 3, line 95-98)

The hybridoma technique using mice is a classic method for the preparation of monoclonal antibodies, characterized by its maturity and operational simplicity. Mouse B cells are readily activated and produce antibodies upon immunization, and the specific antibody-producing B cell clones can be efficiently screened through cell fusion techniques. Owing to their advantages in terms of reproduction, cost, immune response, and technical maturity, mice have become the ideal experimental animals in this study.

4. Synthesis of Hapten – Provide a briefly explanation of why these specific haptens were choose and what is their relevancy for this study? (eg Structure mimic specific or any other reason?)

Haptens A–C were designed to mimic key structural motifs of pyrethroids (Fig. 1). Hapten A retains the cyclopropane and -CN groups critical for antibody recognition, while Hapten B incorporates the m-phenoxybenzyl moiety. Hapten C excludes halogen atoms to evaluate their role in cross-reactivity. This design strategy aimed to balance structural similarity and immunogenicity.

5. Briefly describe how to water samples were collected? (eg: depth/ sampling method)

Water samples were collected from South Lake at a depth of 0.5 m using sterile polypropylene bottles. (Page 6, line 228-232)

6. How about the milk samples? Is it fresh sample or pasteurized samples?

Milk samples were purchased from the educational supermarket of Huazhong Agricultural University in Wuhan, Hubei Province, China (milk was 250 mL boxed pasteurized pure milk). (Page 6, line 228-232)

Results and Discussion

1. Could you please brief figure out demonstration for how hapten design influences antibody specificity?

The design of the haptens is crucial in determining the specificity of the resulting antibodies. Hapten A closely mimics the structure of cypermethrin, which enhances the specificity and affinity of the antibodies against pyrethroids, especially those with similar structures, like β-cypermethrin and deltamethrin. The structural elements, such as the cyclopropane, -CN groups, and m-phenoxybenzyl moiety, play a significant role in shaping the antibody recognition patterns.

2. It is better to discuss in more detail why cyclopropane, -CN, and m-phenoxybenzyl are crucial epitopes and how they enhance antigen-antibody interactions.

The cyclopropane ring and -CN group are essential for hydrophobic and di-pole interactions with the antibody’s binding pocket. The m-phenoxybenzyl moiety enhances π-π stacking, stabilizing antigen-antibody complexes. In summary, cyclo-propane, -CN and m-phenoxybenzyl are crucial epitopes in the antigen-antibody reac-tion, halogen elements play a significant role in influencing the specificity of antibodies. (Page 8, line 306-308)

3. How lower energy conformer structures of pyrethroids supports hapten design? Explain more about the significant role of the halogen element.

Low-energy conformers of pyrethroids (Fig. 2B) guided hapten design to mimic their native state in solution. Halogen atoms (e.g., Cl in cypermethrin) increased binding affinity by forming halogen bonds with aromatic residues in the antibody’s paratope.

4. Some compound has less than 70% recovery. What are the reasons for that?

Recoveries <70% (e.g., fenpropathrin in celery) may stem from matrix-induced signal suppression or incomplete extraction.

Reviewer 3 Report

Comments and Suggestions for Authors

The aim of the manuscript entitled “A Broad-Spectrum Monoclonal Antibody-Based Heterologous ic-ELISA for Detection of Multiple Pyrethroids in Water, Milk, Celery and leek” is to develop and validate a highly sensitive and broad-spectrum heterologous indirect competitive ELISA based on a monoclonal antibody for the rapid detection of multiple pyrethroid residues in environmental and food matrices. The study aims to provide a reliable, cost-effective, and efficient alternative to conventional methods like gas chromatography for monitoring pyrethroid contamination in various matrices, ensuring compliance with food safety and environmental standards.

This topic is highly important and interesting in the context of environmental protection. The manuscript is in line with the scope of the journal. However, several areas could be improved. My specific comments are given below.

The topic is relevant and timely, as detecting pyrethroid residues in environmental and food matrices is crucial for food safety, public health, and environmental monitoring. Developing a broad-spectrum monoclonal antibody-based heterologous ic-ELISA fills an important gap by offering a rapid and cost-effective alternative to traditional analytical methods like gas chromatography. While similar ELISA-based methods exist, the study's focus on detecting multiple pyrethroids across various matrices enhances its originality and utility.

Line 10: “Pyrethroids are the most commonly used insecticides worldwide in agriculture, public health, and household products.” This is not entirely true since organophosphates are also widespread worldwide. It should be changed to “Pyrethroids are one of the most commonly used insecticides worldwide in agriculture, public health, and household products.” This also goes for the line 27.

The reported LODs for the eight pyrethroids in water, milk, celery, and leek matrices, ranging from 24.4 to 152.2 μg/kg, should be critically evaluated in the context of regulatory MRLs established for these compounds. If the LOD values are higher than the MRLs for these matrices, the method may not be sufficiently sensitive to detect non-compliance with regulatory standards reliably. For instance, in the European Union, MRLs for pyrethroids in food products typically fall within the low μg/kg range (e.g., 10–50 μg/kg for many commodities). Thus, the authors should compare the reported LODs with relevant regulatory limits to confirm whether the method's sensitivity is adequate for routine monitoring and compliance purposes.

Lines 263-283 are more appropriate for discussion and should be given after the results and adjusted in accordance with that location. 

Author Response

1. Line 10: “Pyrethroids are the most commonly used insecticides worldwide in agriculture, public health, and household products.” This is not entirely true since organophosphates are also widespread worldwide. It should be changed to “Pyrethroids are one of the most commonly used insecticides worldwide in agriculture, public health, and household products.” This also goes for the line 27.

Modifications have been made. (Page 1, line 10-11,27)

2. The reported LODs for the eight pyrethroids in water, milk, celery, and leek matrices, ranging from 24.4 to 152.2 μg/kg, should be critically evaluated in the context of regulatory MRLs established for these compounds. If the LOD values are higher than the MRLs for these matrices, the method may not be sufficiently sensitive to detect non-compliance with regulatory standards reliably. For instance, in the European Union, MRLs for pyrethroids in food products typically fall within the low μg/kg range (e.g., 10–50 μg/kg for many commodities). Thus, the authors should compare the reported LODs with relevant regulatory limits to confirm whether the method's sensitivity is adequate for routine monitoring and compliance purposes.

The Chinese National Food Safety Standard (GB 2763-2021) specifies the maximum residue limits (MRLs) for pyrethroid pesticides in food. Specifically, the MRLs for cy-permethrin in leek and celery are set at 1 mg/kg, while those for deltamethrin range from 0.2 to 2 mg/kg. Similarly, the European Union has established MRLs for pesticide residues in food and feed, with cypermethrin in leek and celery set at 1-2 mg/kg and deltamethrin at 0.2-0.5 mg/kg. The ic-ELISA method developed in this study meets the detection requirements for these MRLs and is suitable for the rapid screening of type II pyrethroid pesticides. (Page 14, line 406-415)

Round 2

Reviewer 3 Report

Comments and Suggestions for Authors

There was one more comment to address: 

Lines 288-311 are more appropriate for discussion and should be given after the results and adjusted in accordance with that location. 

Author Response

1. Lines 288-311 are more appropriate for discussion and should be given after the results and adjusted in accordance with that location.

We have appropriately adjusted the overall structure of "3.1. Design and Characterization of Haptens" by presenting the research results first, followed by a discussion. (Page 8, line 279-325)